# Antifungal Activity of Ethanolic Extracts from Aeroponically Grown Cape Gooseberry (*Physalis peruviana* L.) with LED Lights and *In Vitro* Habituated Roots

**DOI:** 10.3390/plants13243586

**Published:** 2024-12-23

**Authors:** Daniel Eduardo Avila-Avila, Martha Alicia Rodríguez-Mendiola, Carlos Arias-Castro, Laura Isabel Arias-Rodríguez, Martin Eduardo Avila-Miranda, Norma Alejandra Mancilla-Margalli

**Affiliations:** 1Plant Biotechnology Laboratory, Instrumental Analysis Laboratory, Plant Biochemistry Laboratory, National Technological Institute of Mexico, Tlajomulco de Zuñiga 45640, Mexico; martin.am@tlajomulco.tecnm.mx (M.E.A.-M.); norma.mm@tlajomulco.tecnm.mx (N.A.M.-M.); 2Smart Biotechnology S.A. de C.V., Santa Anita, Tlaquepaque 45600, Mexico; lauraardz@gmail.com

**Keywords:** antifungal activity, Cape gooseberry, indoor farming

## Abstract

Green mold caused by *Penicillium digitatum* is a major post-harvest disease in citrus fruits. Therefore, the search for sustainable and low-environmental-impact alternatives for the management of these fungi is of utmost importance. *Physalis peruviana* L. is a native fruit of the Peruvian Andes with rich bioactive components present throughout the plant. Its antifungal activity stands out, attributed to its high content of phenols, coupled with its antioxidant capacity and antimicrobial activity. Plants were cultivated aeroponically under a combination of red, mixed (50% red, 50% blue), and green LED lights. Additionally, *in vitro*-habituated roots free of plant growth regulators were also cultivated. An ethanol extraction assisted by ultrasound for 30 min followed by maceration for 72 h was performed, and the extract was filtrated and evaporated in an extraction hood. Antioxidant activity was assessed using the DPPH method, total polyphenols were measured using the Folin–Ciocâlteu method, and an antifungal test in vitro by the poisoned food method was conducted against *P. digitatum*. *In vitro* assays revealed that extracts from leaves, roots, and fruits exerted a significant inhibitory effect on the growth of *P. digitatum*, as evidenced by a reduction in colony radius when cultured employing the poisoned food method, with IC_50_ values of 62.17, 53.15, and 286.34 µg·mL^−1^, respectively, compared to 2297 µg·mL^−1^ for the commercial fungicide Captan 50WP. Although leaves had higher total polyphenol content, no direct correlation with antifungal activity was found. Colored LEDs enhanced phenol accumulation, antioxidant capacity, and antifungal properties in plant parts compared to white LEDs and in vitro roots. These findings suggest *P. peruviana* as a new alternative biological production system to provide natural compounds for post-harvest disease management.

## 1. Introduction

Postharvest diseases in citrus fruits caused by fungi can lead to significant losses of up to 50% of global production. Synthetic fungicides are commonly used to prevent these losses, but their effectiveness is limited, with only a 5–10% improvement in total production [1,2]. Among the various fungal pathogens affecting citrus fruits, *Penicillium digitatum* is a major concern, causing green mold and impacting fruit quality and marketability at various stages of the production chain [3,4,5]. Although synthetic fungicides are currently the primary control method, the emergence of resistant strains has reduced their efficacy [6,7]. Plant extracts derived from various plant sources offer a promising alternative to synthetic fungicides due to their natural origin, biodegradability, and potential for reduced environmental impact [8,9]. These extracts contain a diverse array of bioactive compounds, including phenolic compounds, terpenes, and sulfur compounds, which have been shown to exhibit antifungal properties [10,11].

*Uchuva*, also known as Cape gooseberry and golden berry (*Physalis peruviana* L.), a native fruit of the Peruvian Andes, offers numerous health benefits, including anticancer and antimicrobial properties. Its extracts have shown potential in inhibiting fungal growth, particularly against *Candida albicans* and *Aspergillus niger*, attributed to the presence of phenolic compounds and flavonoids. Beyond its direct health benefits, uchuva has emerged as a promising natural resource for controlling plant diseases. Recent research has highlighted its potential in managing post-harvest diseases in citrus fruits, such as green mold caused by *Penicillium digitatum*. The antifungal activity of uchuva extracts is likely due to the presence of phenolic compounds and flavonoids, which can disrupt fungal growth and development. This natural approach could mitigate the risk of fungicide resistance, a growing concern in agriculture. Several studies have confirmed the efficacy of Cape gooseberry extracts in inhibiting the growth of *P. digitatum* [12,13,14].

The use of aeroponic production systems in a controlled environment suggests an optimization of space, water, and nutrient use. Various methodologies have been employed that manipulate environmental stress factors, such as the modification of available nutrients, relative humidity, water stress, plant growth regulators, and light conditions (wavelength, photoperiod, intensity, and light quality). These manipulations induce alterations in physiological processes [15]. Therefore, the quantity of secondary metabolites of the same plant in different areas varies due to the lighting conditions provided. LED lighting has high efficiency in the use of electrical energy, low heat emission, and high manipulation of the intensity and quality of the emitted light. This technology has been employed in indoor cultivation also for its modulatory effect on enzymes involved in the synthesis pathways of secondary metabolites, which are inhibited or activated depending on the intensity, quality, and wavelength (understood as the color of the light) to which they are exposed. The use of LEDs in agriculture is an innovative and efficient alternative to improve metabolic and phenological processes in which light interferes, as they can emit different wavelengths that affect each plant species differently. It has been reported that stimulation with blue and red lights increased the content of polyphenols and antioxidant capacity in carrots by 45% [16]. Furthermore, it has been reported that individually or combined, these lights increase the production of malonyl-CoA, associated with the synthesis of phenolic compounds [17,18]. Blue and red light significantly influence secondary metabolite production in plants by modulating transcriptional and metabolic pathways. This light quality enhances nutrient absorption and assimilation, ultimately improving plant productivity and the nutritional quality of horticultural products, enhancing flavonoid and isoflavone levels. Red light promotes higher kaempferol glycoside content, while blue light stimulates specific isoflavone accumulation [19,20]. There is no literature on the use of aeroponics or the effect of LED lights in the cultivation of *P. peruviana.*

*In vitro* tissue culture is a biotechnological technique that allows us to produce not only disease-free plants but also organs and cells, with various applications such as mass plant production, genetic material preservation, and the exploration of tissues and cells for various biological applications. To be successful with this technique, it is necessary to consider environmental conditions, culture media, plant hormones, and the aseptic conditions of the work area. With this technique, different types of differentiated tissues and undifferentiated cells can also be derived with the aim of obtaining cultures with *in vitro* production of secondary metabolites for different applications [21,22]. Habituation refers to the adaptation of plant tissues to a constant stimulus that can be chemical, physical, or biological and is also mitotically transmissible [23]. Currently, information on the mechanism of habituation is scarce.

This article aims to expand the use of innovative indoor cultivation techniques and the utilization of LED lights to increase the production of secondary metabolites for their application as alternative agents to synthetic fungicides.

## 2. Results and Discussion

### 2.1. Antioxidant Activity

In the antioxidant activity assay by DPPH at a concentration of 5 mg·mL^−1^ (Figure 1), all extracts except for the leaves from treatment 2 and a fruit showed statistically significant differences. The roots demonstrated higher antioxidant activity than the leaves, with the roots treated with colored LED lights exhibiting superior free radical inhibition by DPPH compared to the *in vitro*-habituated roots.

### 2.2. Total Polyphenol Content

The leaves, under both LED light treatments, exhibited highest total polyphenol content (Figure 2). There were no statistically significant differences between the roots and the fruit, but both differed significantly from the leaves, indicating higher polyphenol content per 100 mg of dry weight in leaves. The quantification of total polyphenols represents the sum of all phenolic compounds present, providing an overall view of the total content in our extracts.

### 2.3. Quantification of Quercetin and Apigenin on HPLC

Table 1 displays how different light treatments affected the levels of quercetin and apigenin in *P. peruviana*. White LED light treatment resulted in low levels of both flavonoids in the roots, while colored LED light eliminated quercetin, but significantly increased apigenin. *In vitro*-habituated roots had moderate levels of quercetin and high levels of apigenin, indicating that these conditions favor their production. The leaves had low levels of flavonoids, with a bit more apigenin under colored light. Significant differences in concentrations suggest that light influences the chemical composition of the plant, which is relevant for agriculture and pharmacology.

### 2.4. Antifungal Activity of Extracts in Penicillium digitatum

The ethanolic extracts of *P. peruviana* demonstrated significant antifungal activity against *P. digitatum* using the poisoned food technique (Figure 3). The extracts, particularly from leaves and roots treated with light nutrition and fruit, achieved an average inhibition of 86%, outperforming Captan 50 WP (Figure 4). However, Captan 50 achieved 100% inhibition (Figure 4) at the recommended concentration (2000 µg·mL^−1^). In Table 2, the results of the average inhibition of fungal growth are shown. The lowest IC_50_ values were observed for the fruit (62.18 µg·mL^−1^), root (53.15 µg·mL^−1^), and leaf (286.34 µg·mL^−1^) treated with colored LED lights, indicating a median inhibitory concentration superior to that of the commercial fungicide Captan 50 (2297.32 µg·mL^−1^).

Figure 5 shows the antifungal effect of the extracts evaluated by the poisoned food method in petri dishes with medium, there were significant differences in the fruit and root extracts from the aeroponic system in mixed light compared to white light and control.

### 2.5. Correlation of Antifungal Activity with Total Polyphenol Content

The Pearson correlation analysis (Table 3) revealed a correlation coefficient of antifungal activity to total polyphenol content of −0.293, with a *p* value of 0.165, which is higher than the significance level of *p ≤* 0.01. This suggests that there is no significant relationship between the two variables, indicating that the total polyphenol content in *P. peruviana* is not responsible for the antifungal activity against *P. digitatum*.

## 3. Materials and Methods

### 3.1. Media and Plant Culture

Seed of the Colombian ecotype of *P. peruviana* were used. The seeds were disinfected with an 80% sodium hypochlorite solution (Cloralex^®^, Monterrey, Nuevo Leon, Mexico) in sterilized water with 2 drops of Polysorbate 20, constantly agitated for 20 min, and then triple-rinsed with sterile distilled water. The seeds were sown in solid Murashige and Skoog medium supplemented with 30 g·L^−1^ of sucrose and 2.2 g·L^−1^ of Gelrite and maintained at 25 °C until germination and subsequent development. The PAR light utilized with LED white was 70 µmoles·m^−2^·s^−1^.

### 3.2. Establishment of the Aeroponic Culture

Vitroplants at 30 days after germination (DAG), were transferred to an aeroponic system at 25 °C, with relative humidity maintained between 50% and 60%. A misting regimen was established at 1 min every hour, utilizing a nutrient solution in Table 4. For micronutrients, Ultra Sol^®^ Micro Rexene^®^ BSP Mix (SQM Comercial, Jalisco, Mexico) was used. Two distinct LED light treatments were applied: treatment 1 utilized white LED (intensity 140 µmoles·m^−2^·s^−1^) PAR light, while treatment 2 employed colored LED lights in a configuration of 8 red lamps, 3 mixed (comprising 50% red and 50% blue), and 1 green (all lamps TIANLAI^®^ México City, Mexico), all calibrated to deliver an equivalent photon flux density in µmoles to that of the white LED light. A photoperiod of 16 h of light and 8 h of darkness was established. The plants were exposed to these conditions for 60 days (Figure 6). A combination of red and blue lights was utilized based on prior findings (not presented in this research), indicating that red and blue monochromatic LED lights separately produced significant differences in the antioxidant capacity, as measured by DPPH, in ethanolic extracts of dry leaves of *P. peruviana* cultivated in aeroponic systems. Consequently, the study explored the effect of combining different wavelengths.

### 3.3. Establishment of Habituated In Vitro Root Culture

Root cultures were initiated from 45-day-old *in vitro*-cultured plants in Murashige and Skoog solid medium with 30 g·L^−1^ of sucrose and 2.2 g·L^−1^ of Gelrite without exogenous growth regulators. Under sterile conditions, root segments were excised and placed in Murashige and Skoog solid medium devoid of exogenous growth regulators. These cultures were then incubated in darkness for 15 days at 25 °C. Following this period, the roots were transferred to a liquid MS medium (lacking a gelling agent) and subjected to orbital shaking at 100 rpm. Illumination was provided by white LED lights for a 16/8 h photoperiod, maintaining a constant temperature of 25 °C (Figure 7).

### 3.4. Processing of Plant Samples and Extraction

Extracts were prepared from plant organs (leaves, roots) grown aeroponically and harvested 60 days after transplant (ddt), which involved the cutting of leaves and roots before the formation of floral buds. The *in vitro*-habituated roots were collected at 45 days old. Fruits were obtained from another aeroponic culture grown under white LED light in the same conditions used in the treatments. These were initially frozen at −40 °C in a REVCO freezer (Bunzl Distributioin Inc., St. Louis, MO, USA) and then lyophilized using a Labconco^®^ freeze-dryer (Kanssas City, MO, USA). The dried samples were ground into a fine powder. Extractions were carried out by combining 1 g of the dry sample with 50 mL of 96° G.L ethanol, followed by sonication for 30 min. Ethanol was used as a solvent due to its ability to extract and dissolve secondary metabolites such as phenols, alkaloids, saponins, and terpenes, as well as its low cost and easy accessibility. The samples were then left to macerate at 25 °C for 72 h in darkness. Post-maceration, the mixture was centrifuged and filtered through filter paper using a Büchner funnel and a Büchner flask with vacuum pump. Evaporation was conducted under an extraction hood at room temperature 25 ± 2 °C until the solvent was completely evaporated. The resultant extracts were stored at −4 °C for future use.

### 3.5. Determination Antioxidant Activity by DPPH

Following the methodology outlined in [24], samples were analyzed using a spectrophotometer at 515 nm. The results are expressed as the percentage of DPPH radical inhibition, calculated using Equation (1):(1)$$\mathrm{Antioxidant}\;\mathrm{activity}\;\mathrm{DPPH}\;\left(\%\right)=\frac{\mathrm{Abs}\;515\;\mathrm{nm}\;\mathrm{control}-\mathrm{Abs}\;515\;\mathrm{nm}\;\mathrm{sample}}{\mathrm{Abs}\;515\;\mathrm{nm}\;\mathrm{control}}\times 100$$

Additionally, the calibration curve was constructed using the Trolox standard (0, 20, 60, 100, 120, 150 µg·mL^−1^) to quantify the inhibition in terms of Trolox equivalent antioxidant capacity (TEAC) (µg·mL^−1^), based on the graph’s equation.

### 3.6. Determination of Total Polyphenols by Folin–Ciocâlteu Method

The total polyphenol content was determined using the Folin–Ciocâlteu method as per the methodology of [25]. The calibration curve was established using gallic acid at concentrations ranging from 0 to 1 mg·mL^−1^. The graph’s equation and the coefficient of determination R^2^ were ascertained.

### 3.7. Determination of Antifungal Activity

The strain, provided by the Phytopathology Laboratory of the Technological Institute of Tlajomulco, was cultured in potato dextrose agar (PDA) medium. An antifungal assay was performed using the poisoned food method [26] with 20 mL of medium supplemented with 1 mL of extract. Cylinders of 4 mm diameter were utilized, and four replicates for each treatment were assessed at 3 days post-inoculation. The treatments included ethanolic extracts resuspended in dimethyl sulfoxide (DMSO), subsequently filtered through Whatman^®^ (Kent, UK) NYL w/GMF with a pore size of 0.45 µm by Cytiva Marlborough, MA, USA, fruit (F), leaf in white light (LW), leaf in mixed light (LM), root in white light (RW), root in mixed light (RM), and *in vitro* habituated root (RIV), at dilutions of 10,000, 1000, 100, and 10 µg·mL^−1^. Controls included DMSO (-) and a positive control with Captan 50 WP^®^ at concentrations of 20,000 (recommended concentration), 200, and 20 µg·mL^−1^. Five days after inoculation, the growth radius was measured, the percentage of inhibition was calculated by adjusting for the DMSO control, and the median inhibitory concentration (IC_50_) was determined using the provided equation for each treatment.

### 3.8. Quantification of Quercetin and Apigenin on HPLC

A Thermo Fischer^®^ HPLC system was employed, comprising high-performance anion-exchange chromatography coupled with a pulsed amperometry detector and a diode array detector (DAD). The system utilized a Gemini-NX C18 column (150 mm × 4.60 mm) from Phenomenex^®^ (Torrance, CA, USA). Elution occurred within a gradient system using (A) water acidified with acetic acid (97:3) and (B) acetonitrile. The elution process began with 10% acetonitrile and gradually increased to 90% over a period of 2 to 15 min. The injection volume was 15 µL with a flow rate of 1 mL·min^−1^, while the column temperature remained constant at 30 °C. Two wavelengths (337 nm and 363 nm) were employed for flavonoid detection. Calibration curves were prepared using 95% apigenin and quercetin standards (Sigma-Aldrich, Taufkirchen, Germany) dissolved in HPLC-grade MeOH at six concentrations (µM): 100, 50, 25, 12.5, 6.25, and 0. The hydrolyzed extracts were then diluted to 125 μg·mL^−1^ using the same MeOH.

### 3.9. Correlation of Total Polyphenol Content with Antioxidant/Antifungal Activity

The data underwent Pearson correlation analysis at a significance level of *p* = 0.05. The interpretation of the results is presented in Table 5 [27].

### 3.10. Statistical Analysis

The data underwent one-way ANOVA, followed by an LSD test at a significance level of *p* ≤ 0.05. Pearson correlations were calculated to assess the relationship between the percentage inhibition of *P. digitatum* growth and the total polyphenol content.

## 4. Discussion

The importance of measuring antioxidant capacity lies in the fact that various plant species with antioxidant activity are also associated with other biological activities, such as antifungal activity. Antioxidant compounds with this activity have been reported in these species against different phytopathogenic fungi [28]. While red and yellow light have been shown to stimulate the accumulation of antioxidant compounds in lettuce without significant morphological changes in other species, these lights influence the accumulation of various secondary metabolites like flavonoids, terpenes, and alkaloids [29]. The antioxidant capacity is often attributed to high levels of polyphenols and monoterpenes [30]. However, contrary to these findings, the antioxidant properties in Cape gooseberry roots may not be solely due to polyphenols but could involve a range of secondary metabolites. This study suggests that roots habituated *in vitro* and those grown aeroponically under LED light can synthesize greater quantities of antioxidant compounds compared to other treatments evaluated. Some authors have reported DPPH values in fruits of 210.82 and 192.51 μmol Trolox·100 g^−1^ fruit, with values like ours, but for every 100 mg of sample used in each extract [31].

Leaves tend to have a higher ratio of total polyphenols compared to fruits and other organs, likely due to their role in defense and protection against various stressors, both biotic and abiotic [32]. Another study assessing the accumulation of total polyphenols in lettuce under sunlight and different LED light conditions found the highest polyphenol accumulation in plants grown under mixed light (50% red, 50% blue) [33]. It was found that in *C. paliurus*, there was a greater accumulation effect of flavonoids in leaves under monochromatic blue and red LED conditions. However, the roots contain other phenolic compounds in comparison to the leaves, and when accumulated in greater proportions in this organ, generate the highest antioxidant activity [34].

Components such as quercetin, kaempferol, apigenin, caffeine, and gallic acid have been identified in the fruits of *P. peruviana* through HPLC analysis [35], According to measurements taken in other studies, apigenin and quercetin have demonstrated antioxidant and antifungal activity against strains such as *C. albicans*, *C. parapsilosis*, *C. neoformans*, *A. fumigatus*, *C. parapsilosis*, *T. rubrum*, *T. cutaneum*, and *T. beigelii* [36,37,38,39]. This suggests that the roots may contain and metabolize compounds that are reflected in other organs of the plant. Furthermore, the wavelength of light might affect secondary metabolism, as quercetin was not detected by HPLC in the treatment with a combination of colored LED lights. Additionally, this study contributes to the understanding that roots habituated in vitro and those grown in aeroponics under LED light can synthesize these compounds in greater quantities compared to other evaluated treatments. Regarding the results obtained, apigenin accumulation benefited from blue light, while quercetin was not found in the root and leaf cultivated under LED mix. This could suggest that the enzymes F3′H: flavonoid 3′-hydroxylase and FLS: flavonol synthase are inactivated, or the enzyme flavone synthase oxidoreductase (enzyme responsible for apigenin production) is enhanced by the conditions of the light combination, as they are considered secondary metabolites that plants synthesize directly to adapt to stress light conditions. Other types of phenols have protective effects against certain diseases, while others contribute to shaping the texture and color of fruit [40,41,42]. A combination of red and blue LEDs has been shown to increase the production of flavonoids, lignin, and artemisinin in *Artemisia annua* by 44% and 53.4%, respectively. Red and blue LED lighting significantly increased secondary metabolites, including rosmarinic acid [43]. Short-term exposure to blue light significantly increased total phenols and flavonoids in *Artemisia argyi*, with the highest total phenol content observed at 0.25 mg g^−1^ FW. Blue light also promoted the accumulation of jaceosidin, eupatilin, and taxol [44]. Root exposure to LED light significantly increases secondary metabolites in plants. In *Artemisia annua*, artemisinin accumulation in shoots rose by 2.3- to 2.5-fold, while in *Hypericum perforatum*, coumaroylquinic acid in leaves increased by 89% with various light wavelengths [45]. The applications of LED lights vary according to needs and interests, which can open independent scenarios for each type of problem and tailor the environment to improve the production of bioactive compounds in plants.

In the growth inhibition activity of *P. digitatum*, extracts of *P. peruviana* have not been reported as growth inhibitors of the same. However, the oils from cinnamon leaf and bark, clove, and oregano inhibited 90% to 100% of the fungus development [42]. In other research, inhibition of 86% of *Penicillium* sp. growth was achieved with 1 mL of ethanolic extract of agave, from which 100 g of dry sample of *Agave scabra*, Salm-Dyck was used [46]. Extracts from *Mahonia bealei*, *Ficus semicordata*, and *Gnetum montanum* demonstrated complete inhibition of *P. digitatum* at concentrations of 300–1000 µg·mL^−1^ [47]. *Eriobotrya japonica* L. leaf extract demonstrated a minimum inhibitory concentration (MIC) of 0.625 mg·mL^−1^, affecting mycelial growth and spore germination through alterations in cell membrane permeability and energy metabolism [48]. In the present work, it was found that the ethanolic extracts of the fruit with an IC_50_ of 62 µg·mL^−1^ and of the roots grown in an aeroponic system under LED lights with a mostly red color proportion of 53 µg·mL^−1^ presented an IC_50_ in the growth of *P. digitatum* different from the control treatment with fungicide Captan50 (2297 µg mL^−1^), highlighting that for the preparation of the extract, only 1 g of sample was used, contrary to the quantities used in other methodologies.

Polyphenols maintain antibacterial, antiviral, and antifungal activity, contrary to what was found in these results [49]. However, in this study, there was no direct or proportional relationship between total polyphenols and the antifungal activity of the extracts, suggesting that other components within the extracts contribute to this activity, which contradicts reports of a proportional relationship between total polyphenols and antioxidant capacity [50]. It is inferred that the antioxidant and antifungal activities observed in this study are not due to the total polyphenol content in the *P. peruviana* extracts, but to other compounds present, such as terpenes, which have been reported as antifungals tested against fungi like *Fusarium oxysporum* and *A. alternata*. Similarly, no correlation was found between total polyphenols and antioxidant activity measured by DPPH. Light also affects the biosynthesis of terpenes. Previous studies have shown that red light promotes an increase in terpene production in plants, involving phytochromes in this synthesis of compounds. This results in a difference in the antifungal activity observed in extracts from plants exposed to different light treatments. Other compounds, such as physalins, whitanolides, and saponins, are considered the primary biochemical barriers against a variety of fungal pathogens. Although the specific mechanisms of the antifungal action of saponins are not fully elucidated, it is believed that the aglycone-sugar structure of saponins combines with the sterols of the pathogen in the cell membrane, leading to the disintegration of the cell membrane and pore formation [51]. These metabolites have been reported with antioxidant and antimicrobial activity in the genus *Physalis*, suggesting that these compounds may also be present in the matrix of the *P. peruviana* extracts used during this research. These compounds exhibit interesting biological activities with pharmacological and agronomic relevance. Further investigation of the activities of these components and explore their utilization in different plant organs is recommended [52,53]. Adopting aeroponic systems can significantly improve water and nutrient efficiency, which is especially beneficial in regions with limited water resources. Training in managing these systems is recommended to maximize yields. Using specific LED lights for plant growth can optimize photosynthesis and crop development. It is advisable to conduct pilot tests to determine the most suitable light configurations for each type of crop. Research indicates that specific LED spectra can enhance the synthesis of bioactive compounds, such as phenolics and carotenoids, known for their health benefits [54]. Furthermore, LEDs are energy-efficient and have a long lifespan, making them suitable for large-scale agricultural applications [55]. The combination of LED illumination with these natural extracts could potentially amplify their antifungal effects, providing a sustainable alternative to synthetic fungicides, which face regulatory scrutiny due to health and environmental concerns [56]. This, the synergistic application of LED technology and plant-derived antifungals presents a promising avenue for enhancing crop protection and food safety [57]. Additionally, it is important to research and develop new mineral and light nutrition formulations for *P. peruviana* that are more effective and easier to apply to different types of crops, adapting to specific needs or issues.

## 5. Conclusions

This study infers that the use of colored LED lights in aeroponic systems facilitated the accumulation of metabolites, distinct from apigenin, which were responsible for the noted antifungal activity. The IC_50_ values for *P. digitatum*, when exposed to the ethanolic extracts of the *P. peruviana* fruit, leaf, and root under colored LED lights, were 53 µg·mL^−1^ and 62 µg·mL^−1^, respectively. These values are significantly lower than the positive control with Captan50 WP, which had an IC_50_ of 2297 µg·mL^−1^. Additionally, the analysis revealed no linear correlation between the total polyphenol content in the *P. peruviana extracts* and their antifungal activity against *P. digitatum*. Nevertheless, it is suggested that the use of aeroponic systems and colored LED lights contributed to enhanced metabolites with antioxidant and antifungal activity in the leaves, roots, and fruits, in contrast to those exposed to white LED light and the *in vitro*-habituated roots as an effective method biotechnological to provide natural compounds for post-harvest disease management. The implementation of these systems can enhance the quality and shelf life of agricultural products, reducing post-harvest losses and increasing supply chain efficiency. This is crucial to meet the growing demand for fresh, high-quality food in global markets. Policies that promote research and development in innovative agricultural technologies can accelerate the transition to more sustainable and profitable production systems. Therefore, a new alternative biological production system has been established.

## Figures and Tables

**Figure 1 plants-13-03586-f001:**
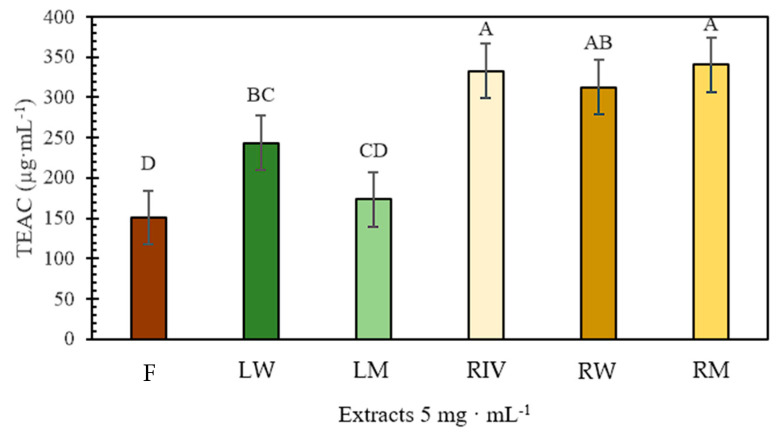
Relationship of Trolox equivalent antioxidant capacity (TEAC) of ethanolic extracts of *P. peruviana* at a concentration of 5 mg·mL^−1^. Fruit (F), leaf in white LED light (LW), leaf mixed LED light (LM), *in vitro*-habituated root (RIV), root white LED light (RW), root mixed light (RM). Statistically significant differences are indicated by different letters LSD, (*p* ≤ 0.05). The error bars in the figure represent the standard error of the mean.

**Figure 2 plants-13-03586-f002:**
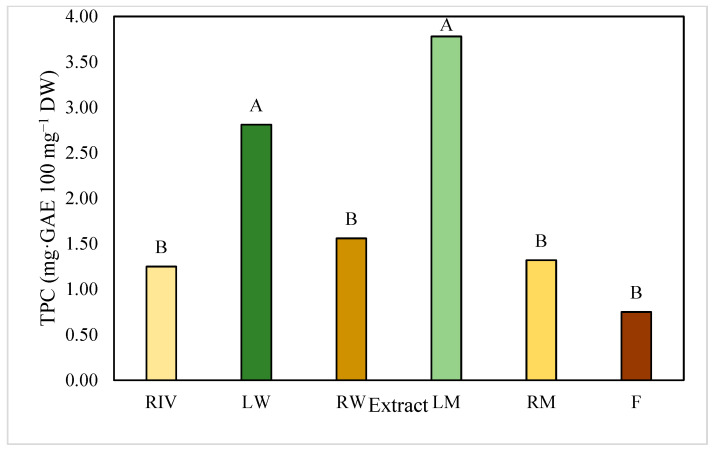
Total polyphenol content (TPC) in extracts of *P. peruviana.* Extracts of *in vitro*-habituated root (RIV), Leaf in white light (LW), Root in white light (RM), Leaf in mixed light (LM), Root in mixed light (RM), Fruit (F). Significant differences are denoted by different letters. LSD (*p* ≤ 0.05).

**Figure 3 plants-13-03586-f003:**
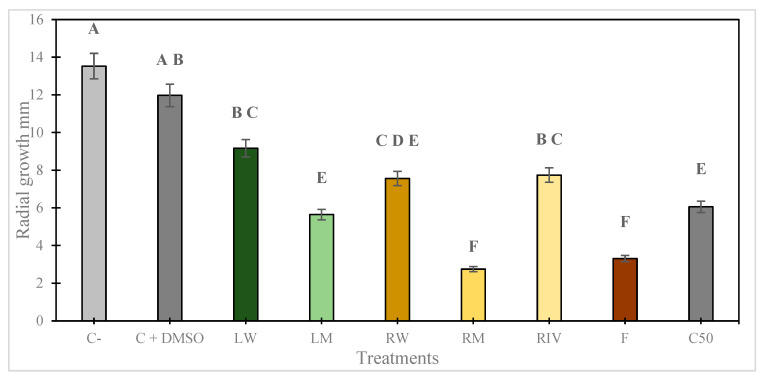
Growth radius in millimeters (mm) of *P. digitatum* in medium poisoned by extracts of *P. peruviana*. Control (C−), control with dimethyl sulfoxide (C+ DMSO), *in vitro*-habituated root (RIV), leaf in white light (LW), root in white light (RW), leaf in mixed light (LM), root in mixed light (RM), fruit (F), Captan 50 WP (C50). The error bars in the figure represent the standard error of the mean. Statistically significant differences are marked by different letters, based on LSD (*p* ≤ 0.05).

**Figure 4 plants-13-03586-f004:**
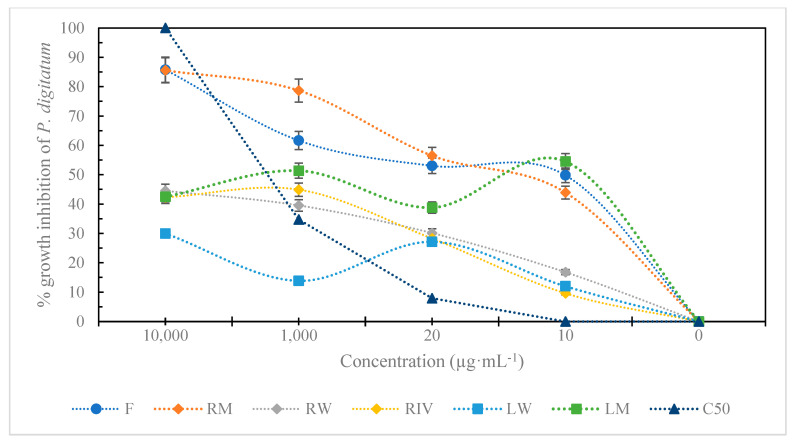
Percentage of growth inhibition of *P. digitatum* using ethanolic extracts of *in vitro*-habituated root (RIV), leaf in white light (LW), root in white light (RW), leaf in mixed light (LM), root in mixed light (RM), fruit (F), Captan 50 WP (C50). The error bars in the figure represent the standard error of the mean.

**Figure 5 plants-13-03586-f005:**
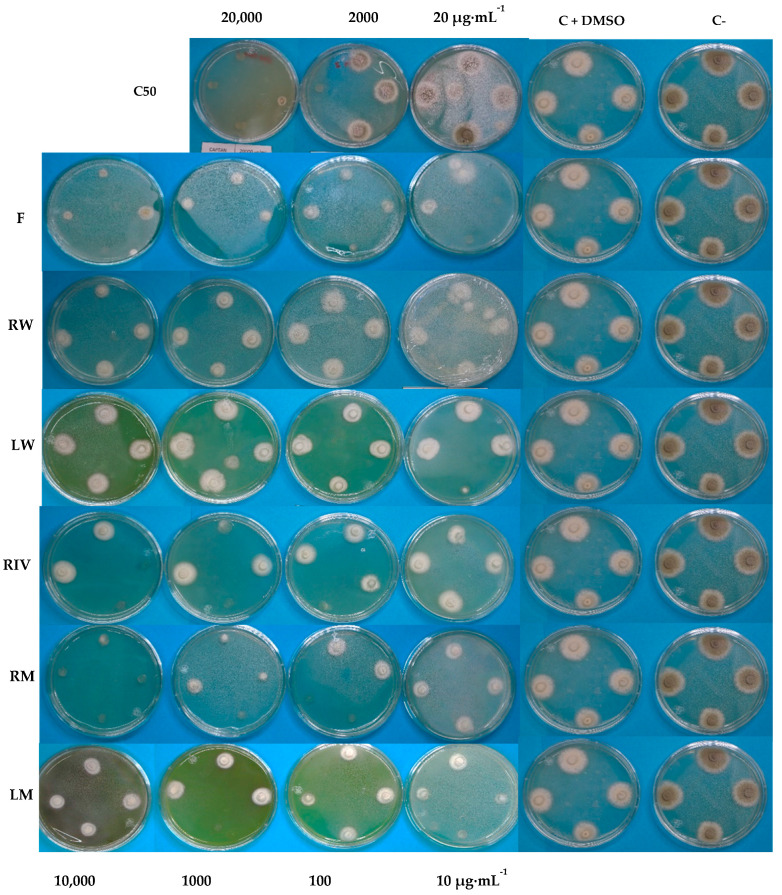
Antifungal activity results in 7 days post-inoculation of *P. peruviana* extracts in *P. digitatum*: Control absolute (C−), control with dimethyl sulfoxide (C + DMSO), positive control with fungicide Captan 50 (C50), fruit (F), leaf white light (LW), leaf mixed light (LM), root in white light (RW), Leaf in mixed light (LM), root in mixed light (RM), *in vitro*-habituated root (RIV).

**Figure 6 plants-13-03586-f006:**
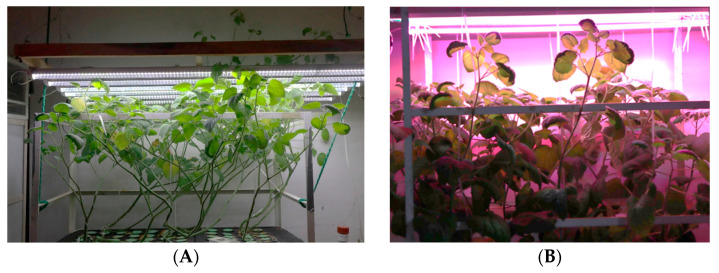
*P. peruviana* cultivated aeroponically: (**A**) cultivated with white LED light and (**B**) with mixed LED light.

**Figure 7 plants-13-03586-f007:**
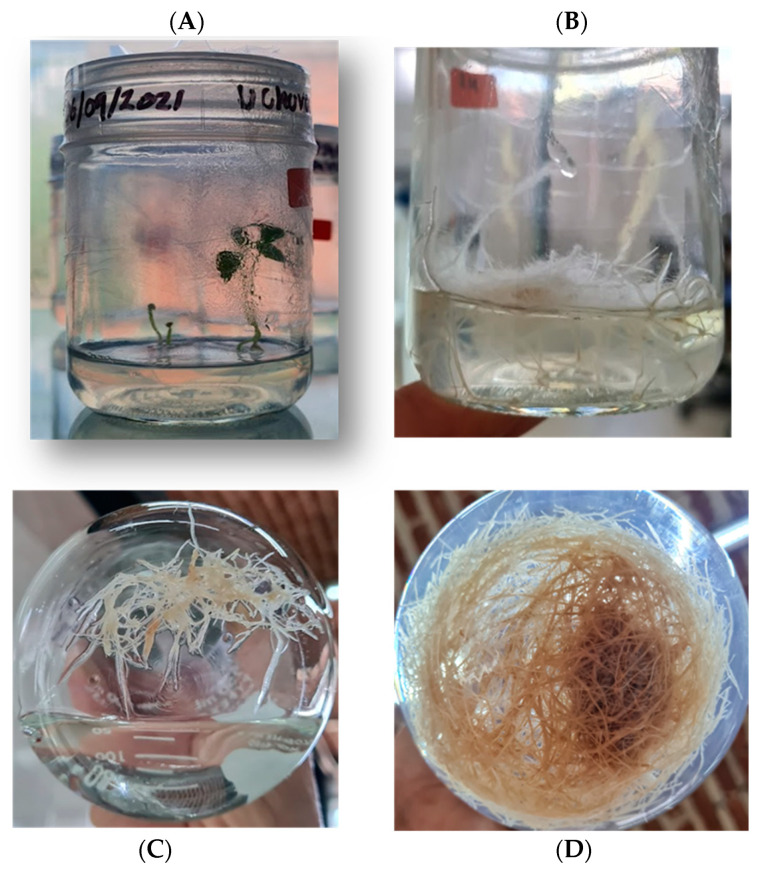
Habituated *in vitro* root culture of *P. peruviana*. (**A**) Germination and growth *in vitro* in solid medium. (**B**) Root growth in solid medium. (**C**) Transfer of roots to liquid medium. (**D**) Habituated roots *in vitro* at 45 days.

**Table 1 plants-13-03586-t001:** Quercetin And Apigenin Content in *P. peruviana* Extracts. *In vitro*-Habituated Root (RIV), Leaf in White Light (LW), Root in white light (RW), Leaf in Mixed light (LM), Root in Mixed Light (RM). Significant Differences are Denoted by Different Letters. LSD (*p* ≤ 0.05).

Extract	Quercetin	Apigenin
	mM ∙ 1 g ^−1^ DW	mM ∙ 1 g ^−1^ DW
RW	0.0002307 **A**	0.0003362 **E**
RM	0 **B**	0.3703847 **A**
RIV	0.04768 **A**	0.361266 **B**
LW	0.009228 **A**	0.00512 **D**
LM	0 **B**	0.00704 **C**

**Table 2 plants-13-03586-t002:** Average inhibitory concentration (IC_50_) of each extract against the *P. digitatum* strain. Root in white light (RW), root in mixed light (RM), fruit (F), Captan 50 WP (C50), *in vitro*-habituated root (RIV), leaf in white light (LW), leaf in mixed light (LM).

Treatment	IC_50_ (µg·mL^−1^)	Equation	Determination Coefficient
RW	1702.15	y = −2 × 10^−6^ x^2^ + 0.0198 x + 22.092	R^2^ = 0.8495
RM	53.15	y = −3 × 10^−6^ x^2^ + 0.0341 x + 48.106	R^2^ = 0.9592
F	62.18	y = −8 × 10^−7^ x^2^ + 0.0119 x + 50.743	R^2^ = 0.9971
C50	2297.32	y = −5 × 10^−7^ x^2^ + 0.0146 x + 19.098	R^2^ = 1
RIV	1169.55	y = −3 × 10^−6^ x^2^ + 0.0319 x + 16.795	R^2^ = 0.8394
LW	2715.86	y = −1 × 10^−6^ x^2^ + 0.0167 x + 12.021	R^2^ = 0.9997
LM	286.34	y = 1 × 10^−7^ x^2^ + 0.0002 x + 55.249	R^2^ = 0.9797

**Table 3 plants-13-03586-t003:** Pearson correlation of antifungal activity and total polyphenol content of all extracts with *p* > 0.01.

Pearson Correlation					
Value 1	Value 2	N	Correlation *r*	IC 99%	*p* Value
Antifungal activity (%)	Total polyphenol content	24	−0.293	(−0.698, 0.254)	0.165

**Table 4 plants-13-03586-t004:** Nutrient Solution For *P. peruviana*.

Nutrient Solution (meq·L^−1^)	Concentration	NO_3_^−^	H_2_PO_4_^−^	SO_4_^2−^	K^+^	Ca^2+^	Mg^2+^
15	%	0.75	0.15	0.1	0.34	0.42	0.24
	meq·L^−1^	11.25	2.25	1.5	5.1	6.3	3.6

**Table 5 plants-13-03586-t005:** Interpretation of R values obtained through a Pearson correlation.

R	Interpretation
0.90	Very strong negative correlation
0.75	Considerable negative correlation
0.50	Medium negative correlation
0.25	Weak negative correlation
0.10	Very weak negative correlation
0.00	No correlation
0.10	Very weak positive correlation
0.25	Weak positive correlation
0.50	Medium positive correlation
0.75	Considerable positive correlation
0.90	Very strong positive correlation
1.00	Perfect positive correlation

## Data Availability

Dataset available on request from the authors.

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
