# Peer review of "Antifungal Activity of Ethanolic Extracts from Aeroponically Grown Cape Gooseberry (Physalis peruviana L.) with LED Lights and In Vitro Habituated Roots"

_plants, 2024, doi:10.3390/plants13243586_

Round 1
Reviewer 1 Report
Comments and Suggestions for Authors
The manuscript, "Antifungal Activity of Ethanolic Extracts from Aeroponically Grown Cape Gooseberry (Physalis peruviana L.) with LED Lights and Habituated Roots In Vitro," is clear and precise. However, it could benefit from a shorter structure.
Abstract: The abstract covers the main points well, but minor writing issues could be improved. For example, the sentence "According to different studies, compounds with antioxidant capacity have other biological activity" is confusing and could be clarified by indicating what types of biological activities it refers to. Additionally, some details about the methods are missing, which is important in a scientific abstract. For example, the authors should briefly mention the antifungal evaluation methods and the key results in quantitative terms.
Introduction: The introduction establishes the importance of the problem well, but it is somewhat repetitive. The importance of plant extracts as an alternative to synthetic fungicides is mentioned several times, which could be condensed. Additionally, the discussion of the relevance of P. peruviana in agriculture could be expanded beyond previous studies. A brief review of other methods to improve the accumulation of secondary metabolites in plants should also be included, such as using different types of light, elicitors, or modulation through cultivation conditions. It does not specify the type of photosynthetic metabolism, whether they are short-day or long-day plants and factors that should support the experimental design. The modulation of metabolites should be robust in both biological activity and physiology/biochemistry to allow an understanding of the cause and effect of differential metabolite accumulation and, consequently, their activities.
Materials and Methods: The experimental design is relevant, but there are areas where details could be improved. For example: • The section on aeroponic cultivation is clear, but it would be useful to specify the duration of LED treatments more precisely. How many days were the plants exposed to the different light treatments before sample collection? • The source of the LEDs used is not mentioned. This is important for the reproducibility of the experiment. • When using light treatments, the dose (electrons or energy received per unit of time, usually per day) should be specified, considering that this effect is relevant. • How do the authors ensure no photoinhibition or other stress factors? • In the extraction section, the solvent evaporation conditions should be further detailed. "Under a fume hood at room temperature" is vague; for example, it should be specified if reduced pressure was used.
Results and Discussion: The results are well organized and presented, but the discussion is limited. • The negative correlation between polyphenol content and antifungal activity is interesting, but the discussion does not thoroughly explore other possible explanations for the observed antifungal activity. It would be valuable to delve into what other compounds might contribute to this activity beyond polyphenols. What happens in the metabolism, i.e., is there a redirection of carbon to a certain group, and how does this relate to the type of cultivation and light treatments? • In the section on apigenin and quercetin content, the importance of these flavonoids in terms of biological activity should be mentioned, as well as whether their synthesis is coordinated or competitive (one is a flavone, normally synthesized by FNS II and the other is a flavonol synthesized by FLS). It should also be discussed whether the red light ratio potentially impacts apigenin accumulation and whether phytochromes are involved in flavonoid metabolism.
Graphs and Tables: • The figures illustrating the growth and inhibition of P. digitatum are useful, but they could be improved in terms of visual clarity. The legends could be more descriptive, better explaining what each letter in the graphs represents. • The IC50 table is well-structured, but including a more direct comparison with other studies or fungicides would be helpful.
References: The references are well-selected and up-to-date, but in some parts, some citations may not be necessary or could be reorganized to improve the text's flow. For example, in the discussion, several citations seem to interrupt the logical flow of the arguments (such as references [26] and [27], which could be integrated more organically).
Conclusions: The conclusion is clear and concise, but it should be strengthened by more emphatically summarizing the practical impact of the findings and their novelty. For instance, it could be suggested how these results could be applied in agricultural programs or the pest control industry, emphasizing the novelty and impact of your work.
General Writing: There are several areas where the writing could be simplified to improve clarity. Some sentences are too long or complex. I suggest breaking long sentences into shorter ones and using simpler terms where possible. This will ensure that your work is presented in the best possible way. Additionally, there are small grammatical errors that need correction.
Formal Aspects: • Ensure that numerical values are always formatted correctly and consistently (for example, use commas or periods consistently to separate decimals). • Acronyms should be defined the first time they appear (for example, IC50 and DPPH are fine, but make sure other acronyms like LED are defined if they haven't been previously).
Author Response
We deeply appreciate your observations, comments, and suggestions, which have been of great help in improving the writing and clearly conveying our work. Below, we summarize the changes made by area:
Abstract: The suggestions were addressed by adding the methodology used.
Introduction: The proposed suggestions were incorporated, adding the factors of metabolite modification and a brief description of the effect of light on secondary metabolism. Additionally, an additional explanation was included in the discussion section.
Materials and Methods: The suggestions were addressed, and the missing descriptions in the methodologies were added. The extraction was carried out "under an extraction hood at room temperature," without being subjected to any other condition or evaporation equipment. The flasks with the extract were left to evaporate in the hood for a considerable time. The LED lamps used were of the TIANLAI brand.
Results and Discussion: The suggested changes were made, adding a discussion on the impact of light on the metabolism of quercetin and apigenin components, as well as on the interaction of light with metabolism.
Figures and Tables: The visual conditions of the figures were improved, and the interpretation of each treatment was changed.
Conclusions: The relevance and suggested applications were added.
General: The writing was improved in some notable points, and the italics were corrected.
Reviewer 2 Report
Comments and Suggestions for Authors
Dear authors,
I have reviewed your manuscript with ID plants-3274955, " Antifungal activity of ethanolic extracts from aeroponically grown cape gooseberry (Physalis peruviana L.) with LED lights and habituated roots in vitro". I appreciate the timely subject of the paper. Overall, the paper is well written but I have a few observations.
Keywords: words that already exist in the title should be changed and written alphabetically.
At its first appearance in the manuscript, write the full Latin name, (Physalis peruviana L.). For subsequent occurrences in the text, use the abbreviated version, P. peruviana (without L)
Please make sure that it is written in italics in the entire text. I noticed that it is not in italics: figure 2, figure 5, line 204, table 2, line 216, figure 7 ...... etc.
Chapter 2.1. Media and Plant Culture – Please elaborate further on this chapter so that the experiment can be reproduced. Were the seeds sterilized? Was the culture medium solid? How many grams of agar were used? etc...
Chapter 2.2. Establishment of the Aeroponic Culture – Please specify the composition of the nutrient solution used.
Please revise the title of Figure 2, as there is a discrepancy between the labels on the images and those in the descriptions.
Lines 118-128 describe the preparation of the extract and should not be included in Chapter 2.3. Establishment of Habituated In Vitro Roots Culture. You could create a separate chapter or distribute the text within each relevant chapter where the plant material was obtained.
In the Results section, please review the statistical analysis. It seems there is an issue with the error bars. First, you need to specify whether they represent the standard error or the standard deviation. In Figures 4 and 7, the error bars appear to be those generated by the "Error Bar" command in Excel. After generating the error bars in Excel, use the "More Options" command, and ensure that you select the calculated values for either Stdev or Sterror, depending on which one you used.
Line 190 - Figure 4. Extracts of fruit [F] ...in the figure you have Fruit. For uniformity, use round brackets here as well.
Line 211 - Table 2. – The abbreviations in parentheses do not match the abbreviations in the table (under the "Extract" column). Please ensure consistency.
The assignment of letters in this table, column Quercetin, is inconsistent. The values of 0 should be labeled with A, indicating that they do not contain Quercetin, while the three higher values (0.0002307, 0.04768, and 0.009228) should receive a different letter, such as B, indicating that they are significantly different from zero.
Line 230 - (C- DMSO) replaced by (C + DMSO). Please check all abbreviations.
The manuscript presents valuable findings regarding the antifungal activity of Physalis peruviana.
Thank you for the opportunity to review this work.
Author Response
We deeply appreciate your suggestions, comments, and the time dedicated to reviewing our article.
Keywords: The keywords were addressed and changed.
General: Corrections were made to the writing of scientific names and their abbreviations.
Materials and Methods: The conditions for seed disinfection and establishment were added. A table with the nutrient solution used was included, and the source of micronutrients utilized is mentioned in Chapter 2.2, Line 103. The description of the establishment of in vitro root cultures and the processing of plant samples and extraction was distributed. The discrepancy in Figure 2 was addressed. The meaning of the error bars was added. The use of brackets and parentheses was corrected. Changes were made in the table of apigenin and quercetin content, noting that 0 (B) is statistically different from the amount found in the extracts. All abbreviations were verified.
We appreciate the observations made, which help us to improve the writing and understanding of our work.
Reviewer 3 Report
Comments and Suggestions for Authors
Editor-in-Chief,
Plants
--------------------------------------------------------
The manuscript entitled “Antifungal activity of ethanolic extracts from aeroponically grown cape gooseberry (Physalis peruviana L.) with LED lights and habituated roots in vitro” is
a valuable paper providing useful information on the effect of ethanolic extracts from cape gooseberry on pathogenic fungi. Many studies have confirmed that cape gooseberry extract has antioxidant and antimicrobial activity against Gram-positive and Gram-negative bacteria. Another study reported the nutritional and bioactive properties of cultivated and wild fruits of P. peruviana. Among the antimicrobial compounds present in P. peruviana fruit extract are flavonoids, phenols, and tannins. These molecules act through different mechanisms, for example, flavonoids are effective against several bacterias because they destroy bacterial cell walls. Tannins form irreversible complexes with nucleophilic amino acids in proteins, promoting bacterial death.
The authors constituted that extracts from leaves, roots, and fruits significantly inhibited P. digitatum growth, with IC50 values of 62.17, 53.15, and 286.34 μg·mL-1, respectively, compared to 2297 μg·mL-1, for the commercial fungicide Captan 50WP. Penicillium digitatum is one of the main fungal pathogens of citrus fruits, as it can occur at various points in the production chains, from the orchard to the sales shelf. A valuable contribution of the authors is the confirmation of the effectiveness of the antifungal effects of the plant extracts, that are more economical and do not present toxic effect to mammals. Thanks for that they can be employed in organic agriculture. The authors' use of aeroponics and LED lights is also an innovative way to improve metabolic and phenological processes.
The paper contains all the elements required in this type of scientific work. Errors include the frequent lack of italics in Latin names, incorrect descriptions of figures and tables, and inconsistent references. The English used should be corrected by a native speaker. Notes to authors are below. The work in the submitted scope is interesting and suitable for publication in Plants.
Remarks for the authors:
Title – suitable
Page 1, Line 4……. is…in vitro ……….should be….in vitro
Page 1, Line 4……. is…whose ……….should be…which
Abstract – suitable
Keywords - suitable
1. Introduction - suitable
Pages 2, 11, Line 54, 259……. is…capegooseberry ……….should be….cape gooseberry
Page 2, Line 79……. is…in vitro ……….should be….in vitro
2. Materials and methods – suitable
Page 2, Line 89……. is…seed ……….should be…seeds
Sentence 89-91 to be corrected
Page 3, Line 106……. is…45 day old ……….should be…45-day-old
Latin names should be in italics.
Pages 4,5,6,7,8,9,10, 11, 12, 13, 14, Lines 115-117, 150, 184, 189, 191, 199, 204, 211, 216, 217, 225, 229, 230, 231, 235, 237, 244, 245, 261, 276, 281, 282, 283, 286, 290, 301, 303, 313, 314, 317, 318, 319, 322, 335, 338, 343, 355, 370, 371, 401, 403, 404…..…....is…in vitro… P. Peruviana…P. Digitatum… Penicillium sp…. Capsicum….Physalis…. Aspergillus niger… Penicillium expansum.. Agrobacterium tumefaciens…Datura stramonium (Solanaceae)..
Fusarium spp…. Agave scabra … Fusarium oxysporum f. sp. physali…..
…….should be…..in vitro… P. peruviana….P. digitatum…… Penicillium sp…. Capsicum…..Physalis……Aspergillus niger… Penicillium expansum .…Agrobacterium tumefaciens…Datura stramonium (Solanaceae)… Fusarium spp. .. Agave scabra…
Fusarium oxysporum f. sp. physali.
Page 4, Line 119……. is… the cutting…The in vitro ……….should be…cutting….In vitro
Page 4, Line 123……. is… ground ……….should be…grounded
Page 5, Line 139……. is… A calibration ……….should be…The calibration
Page 5, Lines 148, …....is… DMSO, Subsequently, …….should be…. DMSO, subsequently,
Page 5, Line 139……. is… μm. fruit ……….should be… μm fruit
Page 5, Line 161,162 ….A) …B)..to throw away
Page 5, Line 162 …. is…10% B ……….should be… 10% acetonitrile
Page 6, Line 181 …. is… the fruit ……….should be… the fruit,
Page 5, Lines 173, Table 1…....is…very …….should be…. Very
3. Results – suitable
Page 6, Line 176…....is… Results and Discussion ….should be….. Results
Page 7, Figure 5…....is…(mg·GAE 100 mg DW)….should be….. (mg GAE·100 mg-1 DW)
Page 9, 11, Lines 232, 233, 245 – no spaces
Improve the Figure 7. Description too close to the chart. Where is on the chart : C- DMSO and C+? The chart shows C50 and C+ DMSO. Lack of order on the chart and in the figure caption.
Improve the Figure 8. It's not clear. The caption under the photos must match the description of the photos – C - DMSO and F, LT1, HL2, LIV, RTI, RT2 but in description is C- + DMSO and F, RT1, LIV, RTI, LT2 and two tmes 1000?.
Table 3 - missing in description RIV. Lack of order in descriptions.
4. Discussions – suitable
Page 11, Line 261…....is… In This …….should be….. This
Page 11, Line 263…....is… ome …….should be….. Others ?
Page 11, Line 265…....is… [28][29] …….should be….. [28,29]
Page 11, Line 270…....is… (50% Red 50% Blue) …….should be….. (50% Red, 50% Blue)
Page 11, Line 271…....is… a higher …….should be….. higher
Page 11, Line 274…....is… highest …….should be….. the highest
Page 11, Line 276…....is… [32],this suggests …….should be….. [32]. It suggests
Page 12, Line 291…....is… highlighting …….should be….. highlighted
Page 12, Line 292…....is… 1g …….should be….. 1 g
Page 12, Line 309…....is… organs. [37][38]. …….should be….. organs [37,38].
5. Conclusions – suitable
Page 12, Line 322…....is…stablished …….should be….. established
References – suitable
Standardize the literature according to Plants requirements.
Comments on the Quality of English Language
The English used should be corrected by a native speaker.
Author Response
We appreciate your time and patience in reviewing our work. Thank you very much for your observations, suggestions, and comments; they will help us improve our research.
The following corrections made by you have been addressed.
Page 1, Line 4……. is…in vitro ……….should be….in vitro
Page 1, Line 4……. is…whose ……….should be…which
Pages 2, 11, Line 54, 259……. is…capegooseberry ……….should be….cape gooseberry
Page 2, Line 89……. is…seed ……….should be…seeds
Page 11, Line 265…....is… [28][29] …….should be….. [28,29]
Page 11, Line 270…....is… (50% Red 50% Blue) …….should be….. (50% Red, 50% Blue)
Page 11, Line 271…....is… a higher ……. should be….. higher
Page 11, Line 274…....is… highest ……. should be….. the highest
Page 11, Line 276…....is… [32], this suggests ……. should be….. [32]. It suggests
Page 12, Line 291…....is… highlighting …….should be….. highlighted
Page 12, Line 292…....is… 1g …….should be….. 1 g
Page 12, Line 309…....is… organs. [37][38]. …….should be….. organs [37,38]
Page 12, Line 322…....is…stablished …….should be….. established
Page 3, Line 106……. is…45 day old ……….should be…45-day-old
Sentence 89-91 to be corrected
All Latin names were corrected.
Figures and were corrected: The figure abbreviations were changed and corrected for each figure. Figures 7 and 8 were addressed and improved
Round 2
Reviewer 1 Report
Comments and Suggestions for Authors
This manuscript, titled “Antifungal Activity of Ethanolic Extracts from Aeroponically Grown Cape Gooseberry (Physalis peruviana L.) with LED Lights and Habituated Roots in vitro,” provides an in-depth exploration of the use of extracts from aeroponically grown Physalis peruviana to combat Penicillium digitatum, a significant post-harvest disease affecting citrus fruits. The research innovatively combines LED light treatments, aeroponic systems, and in vitro root culture to evaluate antifungal and antioxidant activities.
Innovative Approach: This study stands out for its novel method of using aeroponics alongside LED light spectra to enhance the production of secondary metabolites, aligning with sustainable agricultural practices.
Comprehensive Methodology: The research is presented with thorough detail, particularly regarding the light treatments, extraction processes, and antifungal assays, enhancing reproducibility and transparency.
Scientific Relevance: The focus on natural antifungal solutions addresses a crucial agricultural issue, particularly in light of rising resistance to synthetic fungicides.
Clear Data Presentation: Figures and tables are well-constructed, effectively supporting the results with clarity and statistical relevance.
Areas for Improvement
Clarity in the Abstract and Introduction: The abstract is informative but could benefit from a clearer structure to succinctly present the methodology, main findings, and implications. Phrases like "whose fruit is considered exotic and has multiple properties, among who’s its antifungal activity stands out" could be refined for coherence. The introduction provides good context but could be streamlined to avoid repetition regarding the known properties and antifungal uses of Physalis peruviana.
Methodological Details: The extraction process is well-documented; however, including specific reasons for choosing ethanol as the solvent would reinforce methodological rigor. While the LED light specifications are adequately described, explaining why specific colors (e.g., red and blue) were selected and their known effects on secondary metabolite synthesis would strengthen the rationale.
Results Interpretation: The results are comprehensive, but the interpretation could be enhanced by explicitly linking findings to practical applications. For example, highlighting the potential scalability of aeroponic systems with LED lights for large-scale agricultural use would provide valuable context. The correlation analysis suggests that total polyphenol content did not correlate significantly with antifungal activity. Expanding on why other compounds, such as terpenes or withanolides, might contribute to the observed effects, supported by additional literature, would be insightful.
Discussion and Broader Implications: The discussion effectively connects the findings with previous research, but it could be enriched by exploring future directions, such as the potential use of LED light treatments in other plant species with antifungal properties. Consider including more specific recommendations for agricultural stakeholders or researchers aiming to apply these findings practically.
Conclusions: The conclusion effectively summarizes the key points, but emphasizing the broader significance of these findings in post-harvest management and potential policy implications would strengthen the final message.
Technical Edits:
Terminology Consistency: Ensuring consistency in terms like “phenolic compounds” and “polyphenols” would improve clarity. Grammatical Refinements: Address minor grammatical improvements for enhanced readability. Formatting: Double-check the consistency of figure and table labels to align with journal guidelines.
Author Response
We appreciate the time you took to review our work.
Added the reasoning behind the use of the light combination was added. A preliminary pilot experiment on the effect of monochromatic LED lights on P. peruviana plants grown in aeroponic systems was actually conducted, but this experiment was only carried out for 30 days of growth. Subsequently, the scaling up of that experiment led to this final research.
Some modifications were made to the abstract to enhance understanding, but to avoid exceeding the word limit, they were kept to a minimum. Information regarding the use of blue and red LED lights in the modification of secondary metabolism was added. Additionally, previous comments were incorporated into the discussion. Added bibliography of the use of LED lights in agriculture and some examples in secondary metabolites.
Comments were also added in the conclusion, including the implications of our work for post-harvest diseases and its scalability.
Reviewer 2 Report
Comments and Suggestions for Authors
Thanks to the Authors for the corrections.
Author Response
Thank you very much for your comments, suggestions dedicated to this work.